# Genetic Linkage between *CAPN5* and *TYR* Variants in the Context of Albinism and Autosomal Dominant Neovascular Inflammatory Vitreoretinopathy Absence: A Case Report

**DOI:** 10.3390/ijms25126442

**Published:** 2024-06-11

**Authors:** Mirjana Bjeloš, Ana Ćurić, Mladen Bušić, Benedict Rak, Biljana Kuzmanović Elabjer

**Affiliations:** 1University Eye Department, Reference Center of the Ministry of Health of the Republic of Croatia for Inherited Retinal Dystrophies, Reference Center of the Ministry of Health of the Republic of Croatia for Pediatric Ophthalmology and Strabismus, University Hospital “Sveti Duh”, 10000 Zagreb, Croatia; dr.mbjelos@gmail.com (M.B.); akrizanovic25@gmail.com (A.Ć.); benedict.rak@gmail.com (B.R.); belabjer@kbsd.hr (B.K.E.); 2Faculty of Medicine, Josip Juraj Strossmayer University of Osijek, 31000 Osijek, Croatia; 3Faculty of Dental Medicine and Health Osijek, Josip Juraj Strossmayer University of Osijek, 31000 Osijek, Croatia

**Keywords:** retinal diseases, ERG, VEP, microperimetry, OCT-A

## Abstract

We present a case involving a patient whose clinical phenotype aligns with oculocutaneous albinism (OCA), yet exhibits a complex genotype primarily characterized by variants of unknown significance (VUS). An 11-year-old boy manifested iris hypopigmentation and translucency, pronounced photophobia, diminished visual acuity and stereopsis, nystagmus, reduced pigmentation of the retina, and foveal hypoplasia. Genetic testing was performed. A heterozygous missense VUS *CAPN5* c.230A>G, p.(Gln77Arg), a heterozygous missense VUS *TYR* c.1307G>C, p.(Gly436Ala), and a heterozygous missense variant *TYR* c.1205G>A, p.(Arg402Gln) which was classified as a risk factor, were identified. We hypothesized that the *TYR* c.1307G>C, p.(Gly436Ala) variant is in genetic disequilibrium with the *TYR* c.1205G>A, p.(Arg402Gln) variant leading to deficient expression of melanogenic enzymes in retinal cells, resulting in the manifestation of mild OCA. Additionally, this study represents the case where we did not detect chiasmal misrouting in visual evoked potentials, nor did we observe a shift in the distribution of ganglion cell thickness from a temporal to a central position. Moreover, our patient’s case supports the probable benign nature of the *CAPN5* c.230A>G, p.(Gln77Arg) variant.

## 1. Introduction

Pigmentation is a polygenic trait influenced by various genetic factors and the quantity and type of pigments [1]. Within the human population, pigmentation demonstrates a wide spectrum of diversity. Oculocutaneous albinism (OCA) manifests as a deficiency or absence of melanin in the skin, hair, and eyes, with non-syndromic OCA typically following an autosomal recessive inheritance pattern [2]. The primary genetic determinants of non-syndromic OCA reside in the *TYR* (OCA1), *OCA2* (OCA2), *TYRP1* (OCA3), and *SLC45A2* (OCA4) genes [3]. The prevalence of all forms of albinism globally is estimated at approximately 1 in 17,000 individuals with OCA1 being the most prevalent form, stemming from mutations in the *TYR* gene on chromosome 11q14.3 [2], encoding tyrosinase—a membrane glycoprotein crucial in melanin production [4,5].

The clinical manifestations of OCA encompass a range of ocular abnormalities, including reduced visual acuity, refractive errors, color vision impairment, iris hypopigmentation and translucency, reduced pigmentation of the retinal pigment epithelium (RPE), photophobia, misrouting of optic nerves, strabismus, reduced stereoscopic vision, congenital nystagmus, and foveal hypoplasia (FH) being the hallmark of albinism [6]. The severity of OCA correlates with a pigmentation threshold within affected individuals. Individuals with higher pigmentation backgrounds, such as those of Black ethnicity, typically require two severe mutations to disrupt pigmentation pathways and exhibit visible phenotypes. Conversely, individuals with lighter complexions, such as Caucasians and Hispanics, may present with a broader spectrum of OCA phenotypes, influenced by the presence of mutations or alleles specific to their ethnic backgrounds [7].

In accordance with melanin biosynthesis, OCA subtypes span from the most severe OCA1A type, characterized by undetectable tyrosinase activity and lifelong absence of melanin synthesis, to milder forms (OCA1B, OCA2, OCA3, and OCA4) where some pigment accumulation over time may mitigate symptoms [6]. On the contrary, in ocular albinism (OA), an X-linked condition, only the eyes are affected. Mutations in *GPR143*, a gene involved in the maturation of macromelanosomes, are the only known cause of OA [8]. Given the clinical overlap between OCA subtypes, molecular diagnosis assumes paramount importance in establishing the precise gene mutation and OCA subtype. 

Located at q14.1, *CAPN5*, responsible for encoding autosomal dominant neovascular inflammatory vitreoretinopathy (ADNIV), shares this chromosome with *TYR*, representing a genomic neighborhood significantly implicated in ocular physiology and pathology. This genomic arrangement underscores the potential genetic and functional relationships between these genes, despite their disparate roles in pigmentation and ocular health, respectively [9,10,11,12].

ADNIV, mediated by *CAPN5* mutations, shares clinical hallmarks with retinitis pigmentosa, uveitis, and proliferative diabetic retinopathy, including ocular inflammation, vascular dysregulation, hyperpigmentation, neovascularization, vitreous hemorrhage, and retinal detachment [13].

In this study, we present a case involving a patient whose clinical phenotype aligns with OA, yet exhibits a complex genotype primarily characterized by variants of unknown significance (VUS), notably including a heterozygous missense variant in the *CAPN5* gene c.230A>G, p.(Gln77Arg)), alongside heterozygous missense variants in the *TYR* gene c.1205G>A, p.(Arg402Gln) and c.1307G>C, p.(Gly436Ala).

## 2. Case Presentation

An 11-year-old boy was referred to our Reference Center for Inherited Retinal Dystrophies for comprehensive ophthalmic examination and genetic testing. The parents noticed that he was closely approaching the television screen while watching it. At the ophthalmological examination in 2023, suboptimal distance visual acuity that could not be improved with glasses was determined, and a flattened macular area was depicted by optical coherence tomography (OCT). In the family, the patient’s maternal grandmother and her brother exhibited a phenotype characterized by complete depigmentation of the skin and hair during childhood, while the patient’s father displayed a transition from light to dark pigmented hair over time. The parents of the patient exhibited good health and normal vision. On clinical examination, marked photophobia and hypopigmentation of the irises but not of the hair and skin was evident. The patient had brown hair and tanned normally. His best-corrected visual acuity (BCVA) tested with Lea Symbols in lines was 0.3 logMAR (decimal score 0.5) tested at 4 m binocularly. Tested at 40 cm binocularly, the BCVA measured 0.0 logMAR (decimal score 1.0). Tested monocularly at 4 m, BCVA was 0.7 logMAR (decimal score 0.2) on the right eye (RE), and 0.3 logMAR on the left eye (LE) (decimal score 0.5); however, when tested at 40 cm, the RE and LE values were 0.2 logMAR (decimal score 0.63) and 0.1 logMAR (decimal score 0.8), respectively. The Lang and Titmus fly test were negative. Nystagmus was consistently observed during monocular viewing, regardless of the distance. During binocular viewing, nystagmus was only present at distance and completely ceased upon convergence when focusing on nearby objects. This phenomenon elucidated the observed differences in BCVA. 

Farnsworth’s D-15 dichotomous test and Lanthony desaturated 15-hue panel were unremarkable. CSV-1000 contrast sensitivity test for the spatial frequencies of 3, 6, 12, and 18 cpd was unremarkable and measured binocularly: 1.78, 2.29, 1.99, and 1.55 log units, while monocularly on the RE: 2.08, 2.14, 1.99, and 1.55 log units, and LE: 1.78, 2.29, 1.99, and 1.55 log units. Biomicroscopy revealed blue irises with translucency grade 1 [8], elucidating pronounced photophobia (Figure 1). No vitreous cells were visible.

Octopus^®^ (Haag-Streit Inc., Mason, OH, USA) G, TOP, SAP, *w*/*w*, III static perimetry, was unremarkable: mean deviation of -0.1 dB on the RE and 0.7 dB on the LE and mean sensitivity of 28.1 dB and 27.3 dB on the RE and LE, respectively, were achieved.

MAIA microperimetry (iCare Finland Oy, Vantaa, Finland) (20°, 68 points) reached an average threshold of 27.1 on the RE and 26.2 on the LE with stable fixation on the RE and relatively unstable fixation on the LE.

Optos^®^ California (Optos Inc., Marlborough, MA, USA) ultra-widefield imaging depicted the optic nerve head with clear boundaries on both eyes (BE), present macular reflexes on BE, with RPE mottling on the LE in the foveal area. The retina was extremely light pigmented (fundus hypopigmentation grade 2) [8]. Fundus autofluorescence depicted a less pronounced hypoautofluorescent pattern in the macular area.

HRA + OCT Spectralis^®^ (Heidelberg Engineering, Heidelberg, Germany) imaging depicted flattened foveal depression on BE, while on the LE, central disruption of photoreceptors and the ellipsoid zone was visible, with a central macular thickness of 321 μm on the RE and 316 μm on the LE (Figure 2). 

Ganglion cell layer thickness (GCLT) was similar between the temporal and nasal areas (Figure 3). 

OCT angiography (OCT-A) showed the absence of a foveal avascular zone (FAZ) on BE.

Retinoscopy demonstrated a refractive error of +1.00 Dsph/−0.75 Dcyl ax 180°on the RE, and +1.25 Dsph/−0.75 Dcyl ax 170 on the LE.

Pattern reversal visual evoked potential (p-VEP) albino protocol 1-channel testing (Roland Consult RETIport/scan 21, Roland Consult Stasche and Finger GmbH–German Engineering, Brandenburg an der Havel, Germany) performed using goldcup electrodes was inconclusive, without a characteristic response for misrouting [14]. 

Full-field electroretinography (FFERG), performed according to the International Society for Clinical Electrophysiology of Vision (ISCEV) standards [15], revealed unremarkable generalized retinal function bilaterally (Figure 4). The ISCEV Standard ERG series defines six protocols, each designated based on the stimulus intensity (measured in cd·s·m^−2^) and the adaptation state of the retina [15]. The names of each protocol reflect the specific testing conditions and responses targeted, ensuring standardized and reproducible assessments of retinal function across different clinical and research settings. The specified recording conditions mandate a 20-min dark-adaptation period before recording dark-adapted ERGs and a 10-min light adaptation period before light-adapted ERG testing. The dark-adapted (DA) ERGs include responses to flash strengths (in photopic units) of 0.01, 3, and 10 cd·s·m^−2^, known as DA 0.01, DA 3.0, and DA 10.0, respectively (Figure 4). The light-adapted (LA) ERGs include responses to a flash strength of 3 cd·s·m^−2^, superimposed on a light-adapting background with a luminance of 30 cd·m^−2^. These are tested as single flashes (LA 3.0 ERG) and at a frequency close to 30 Hz (LA 30 Hz ERG) (Figure 4). The dark-adapted 0.01 ERG protocol measures a rod-driven response primarily mediated by ON bipolar cells. Dark-adapted 3.0 ERG captures combined responses originating from both rod and cone photoreceptors and bipolar cells, with a predominance of rod activity. Dark-adapted 10 ERG provides a combined response with pronounced a-waves indicative of photoreceptor function. Dark-adapted oscillatory potentials (OPs) focus on responses predominantly from amacrine cells. Light-adapted 3.0 ERG assesses cone system responses, where a-waves are derived from cone photoreceptors and cone OFF bipolar cells, while b-waves originate from ON and OFF cone bipolar cells. Light-adapted 30 Hz flicker ERG evaluates a sensitive response driven by the cone pathway.

### Materials and Methods

A saliva sample was collected for genetic testing and sequence analysis using the Blueprint Genetics Retinal Dystrophy Panel Plus (version 7, 30 October 2021) which identified a heterozygous missense VUS *CAPN5* c.230A>G, p.(Gln77Arg), a heterozygous missense VUS *TYR* c.1307G>C, p.(Gly436Ala), and a heterozygous missense variant *TYR* c.1205G>A, p.(Arg402Gln) which was classified as a risk factor. The next-generation sequencing data indicated that the *TYR* variants are on different parental chromosomes (in trans) in this patient.

Variant classification followed the Blueprint Genetics Variant Classification Schemes modified from the American College of Medical Genetics and Genomics guideline 2015 [16,17]. The classification and interpretation of the variants identified reflect the current state of Blueprint Genetics’ understanding at the time of this report. Variant classification and interpretation are subject to professional judgment, and may change for a variety of reasons, including but not limited to, updates in classification guidelines and availability of additional scientific and clinical information.

Genetic testing of the parents was not conducted in this study due to their documented state of health and the absence of a clinical indication warranting such testing.

## 3. Discussion

### 3.1. TYR

Tyrosinase is a membrane glycoprotein crucial in melanin production, encoded by the *TYR* gene [4,5]. The monophenolase and diphenol oxidase activities are linked to the tyrosinase active site, which is composed of six histidine residues that coordinate two copper ions (CuA and CuB) essential for activity [18,19].

#### 3.1.1. NM_000372.5(TYR):c.1205G>A (p.Arg402Gln)

In ClinVar, this variant was identified with conflicting classifications of pathogenicity: other, uncertain significance (4); benign (2); likely benign (3) [20].

The c.1205G>A, p.(Arg402Gln) variant within the *TYR* gene represents a prevalent polymorphism with a global occurrence rate of 17.7%, notably abundant in European Caucasian populations, where it exhibits an allele frequency of 26.48% [21]. While typically considered a neutral polymorphism rather than a pathogenic variant due to its lack of inducing albinism in the homozygous state, its potential involvement in mild forms of OCA has been under scrutiny since 1991 [22]. Overall, OCA1B mutations located in exon 4, among other c.1205G>A, encompass the CuB binding site and may affect the geometry of the substrate binding site or the interaction of the substrate with the binuclear copper site, resulting in reduced tyrosinase activity [23,24]. Genetic changes in this loop may play a role in the thermal sensitivity of human tyrosinase and possibly define the temperature-sensitive phenotypes [25]. This specific variant encodes a thermosensitive tyrosinase, demonstrating only 25% of normal catalytic activity at 37 °C, owing to its sequestration within the endoplasmic reticulum, yet it is liberated at temperatures between 31 °C and 32 °C. Intriguingly, our findings highlight a pathogenic association of this variant in the individual affected by albinism who harbors a solitary mutation in OCA1. This observation is supported by the research of Hutton and Spritz who studied patients with autosomal recessive OCA [26], as well as by the novel hypothesis proposed by Chiang et al. regarding OCA1B [7].

Further, Monferme et al. performed phenotypic analysis of 69 compound heterozygous patients with ocular features of albinism harboring the *TYR* c.1205G>A, p.(Arg402Gln) variant [21]. Most patients presented with white or yellow-white hair at birth (71.43%), blond hair later (46.97%), a light phototype but with residual pigmentation (69.64%), and blue eyes (76.56%). Their pigmentation was significantly higher than in the classical OCA1 group, as was in our patient. The authors concluded that the *TYR* c.1205G>A, p.(Arg402Gln) variant leads to variable but generally mild forms of albinism whose less typical presentation may lead to underdiagnosis [21]. In our case, the OCA phenotypic spectrum was dependent on the presence of a hypomorphic or ethnic-specific allele and a mutation *TYR* c.1307G>C, p.(Gly436Ala).

#### 3.1.2. NM_000372.5(TYR):c.1307G>C (p.Gly436Ala)

To the best of our knowledge, this variant has not been reported in the medical literature. In ClinVar, this alteration is classified as a VUS [27]. The *TYR* c.1307G>C, p.(Gly436Ala) variant has been identified in the gnomAD population database at a frequency of 0.003%, with five individuals being heterozygous. Predictive analyses employing Polyphen and SIFT suggest that the variant is likely to be tolerated. This nucleotide alteration results in the substitution of glycine, a neutral and non-polar amino acid, with alanine, which similarly possesses neutral and non-polar properties. Moreover, advanced modeling of the protein sequence and biophysical properties indicate that this missense variant is not expected to disrupt TYR protein function [27]. 

Although the variant does not meet the classification criteria outlined by the American College of Medical Genetics and Genomics for pathogenicity or likely pathogenicity, our patient unequivocally exhibited characteristic signs and symptoms of OA, including iris hypopigmentation and translucency, pronounced photophobia, diminished BCVA and stereopsis, nystagmus, reduced pigmentation of the RPE, and FH, as documented in the previous literature. This strongly suggests a potential contribution of the variant to the observed phenotype. Additionally, the analysis and interpretation of albinism should be contextualized within the genetic background of patients, emphasizing the importance of considering individualized perspectives that may deviate from strict adherence to established guidelines [7]. In lieu with these findings, our patient manifested a mild form of OCA. This presentation, characterized by normal pigmentation in cutaneous tissues, but hypopigmentation in ocular tissues, suggests the deficient expression of melanogenic enzymes in the ocular tissue rather than an inherent inability to produce melanin [28], prompting further inquiry into the underlying factors contributing to the prominence of ocular involvement only in this particular case. One potential explanation is that the genetic mutations responsible for OCA predominantly affect the expression of melanogenic enzymes in ocular tissues rather than in cutaneous tissues. This differential expression pattern could result from tissue-specific regulatory mechanisms or variations in the penetrance of the mutations within different cell types. The specific genetic variants present in our patient may have a greater impact on melanin production in the eyes compared to the skin, leading to pronounced ocular hypopigmentation while sparing cutaneous tissues. This could stem from differential expression levels or functional effects of the mutated genes in ocular versus cutaneous melanocytes.

### 3.2. CAPN5

The *CAPN5* gene encodes calpain-5, ubiquitously expressed calcium-dependent cysteine proteases functioning in a variety of intracellular processes, but strongly expressed in the photoreceptor cells of the retina [13].

Heterozygous damaging variants in the *CAPN5* gene are associated with ADNIV: an autosomal dominant blinding disorder characterized by ocular inflammation, vitreous hemorrhage, and retinal detachment [13]. The disease onset varies between 10 and 30 years of age; however, most affected individuals are asymptomatic until their third or fourth decade. The course of the disease can be divided into five stages, each lasting approximately ten years. The first stage is clinically indistinguishable from autoimmune, non-infectious vitritis, with selective reduction in the b-wave of the FFERG. The second stage is characterized by retinitis-pigmentosa-like photoreceptor degeneration, far-peripheral arteriolar closure, and pigmentation. The third stage is marked by retinal and iris neovascularizations. The fourth stage involves proliferative vitreoretinopathy leading to retinal detachment, and neovascular glaucoma. The final fifth stage culminates in phthisis and complete blindness [13,29].

#### NM_004055.5(CAPN5):c.230A>G (p.Gln77Arg)

This variant is absent in gnomAD, and according to in silico predictors, it is indicated as pathogenic (BayesDel addAF and BayesDel noAF), benign (DEOGEN2, EVE, Mutation assessor, PROVEAN, SIFT, DANN, EIGEN, M-CAP), and uncertain (MetaLR, MetaRNN, MetaSVM, REVEL, BLOSUM, FATHMM, LIST-S2, LRT, MutationTaster, MutPred, MVP, PrimateAI) [30]. A Grantham score of 43 suggests a conservative substitution between glycine and arginine indicating the likely benign nature of the variant. 

Of particular interest is the observation that this variant has been documented in the literature in a single patient, who exhibited recurrent, bilateral panuveitis during adolescence, without any additional clinical manifestations associated with ADNIV [31]. In our present case, a thorough assessment did not detect any signs or symptoms consistent with ADNIV, including the absence of early b-wave FFERG decline (Figure 4). Further support for the non-pathologic nature of this variant stems from the observed correlation between OCA defects and diminished dopamine production [32], stimulating vascular endothelial growth factor (VEGF) and angiogenesis [33]. On the other hand, *CAPN5*-ADNIV patients were previously found to have elevated VEGF levels in the vitreous, and intravitreal anti-VEGF injection suppresses neovascularization and clears vitreous hemorrhages [34]. Therefore, the simultaneous presence of pathogenic *CAPN5* and *TYR* variants is anticipated to exert a synergistic influence on VEGF activity, potentially resulting in the emergence of the pronounced ADNIV phenotype.

### 3.3. Albinism

The macular pigment appears around 17 weeks’ gestational age and is likely one driving force to the formation of FAZ [35]. Foveal pit formation does not take place until the FAZ starts to form with the morphology of the foveal pit correlating positively with the FAZ area [35]. It is hypothesized that the lack of macular pigment halts FAZ formation which in turn delays foveal pit formation [35]. OCA1 defects all result in lower dopamine production [32], and dopamine inhibits angiogenesis induced by VEGF [33]. Thus, excess VEGF could possibly promote vascularization, impede formation of the FAZ, and contribute to FH [32].

With approximately 16.1% of patients exhibiting no misrouting in VEP tests [8], FH emerges as the predominant clinical manifestation in albinism, observed to be absent in merely 0.7% of cases [36]. Visualization of FH with OCT is considered crucial in guiding the diagnosis of albinism, especially in patients exhibiting milder clinical presentations [37]. According to the Leicester Grading System for FH developed by Thomas et al. [38], our patient presented with grade 2 FH on the RE and atypical FH on the LE. Consistent with the findings of Kuht et al., who noted that OA individuals displayed higher grades of FH (3 and 4), in contrast to OCA individuals who exhibited FH grades ranging from 1 to 4, our patient’s presentation aligned with that of OCA [39]. 

Due to the extensive phenotypic variability observed in albinism, Kruijt et al. proposed a diagnostic criterion wherein, with a molecular diagnosis present, confirmation of albinism requires meeting either one major criterion or two minor criteria [8]. Our patient meets two major criteria, specifically FH grade ≥ 2 and ocular hypopigmentation, thus supporting the diagnosis of albinism.

However, our patient did not exhibit any discernible misrouting in the VEP nor demonstrate the typical distribution of GCLT seen in individuals with albinism, which typically involves a shift of macular ganglion cells from a temporal to a central position [37], potentially explaining the lack of misrouting. Specifically, chiasmal misrouting involves an increased number of retinal ganglion cells from the temporal retina projecting to the contralateral hemisphere, differing from the normal routing pattern where the temporal retina projects to the ipsilateral hemisphere. This association has not been previously documented in the literature.

*TYR* c.1205G>A is strongly associated with patients who have albinism [7] and does not cause autosomal recessive OCA, suggesting that a causative variant may be in genetic disequilibrium with the c.1205G>A variant [40]. Therefore, we hypothesize that the *TYR* c.1307G>C, p.(Gly436Ala) variant could be in genetic disequilibrium with the *TYR* c.1205G>A variant, causing the phenotype expressed by our patient.

## 4. Conclusions

We have elucidated the presence of OCA in a patient harboring unique variants of the *TYR* and *CAPN5* genes, providing compelling evidence for the pathogenicity of *TYR* c.1205G>A, p.(Arg402Gln) when in trans configuration with another pathogenic variant. Remarkably, the patient exhibited a mild OCA phenotype exhibiting ocular manifestations only. 

The *TYR* c.1307G>C, p.(Gly436Ala) variant is classified as a VUS, lacking sufficient evidence to ascertain its pathogenicity. Nonetheless, our patient unequivocally displayed signs and symptoms consistent with OCA, strongly implying the pathogenic contribution of this variant to the observed phenotype.

Moreover, the *TYR* c.1205G>A, p.(Arg402Gln) variant has been robustly associated with albinism in previous studies, with the suggestion that a causative variant might be in genetic linkage disequilibrium with it [40]. Therefore, we hypothesize that the *TYR* c.1307G>C, p.(Gly436Ala) variant is in genetic linkage disequilibrium with the *TYR* c.1205G>A, p.(Arg402Gln) variant, consequently resulting in the manifestation of mild OCA in our patient.

Notably, this study marks the first instance where the misrouting in VEP has been correlated with a shift in the distribution of GCLT, an association hitherto unreported in the existing literature.

The *CAPN5* c.230A>G, p.(Gln77Arg) variant is classified as a VUS due to inadequate evidence to assess its clinical significance. However, our patient did not exhibit any discernible features specific to ADNIV; particularly, the earliest indicators of b-wave decline on FFERG were absent. Notably, our patient’s case supports the probable benign nature of the variant.

## Figures and Tables

**Figure 1 ijms-25-06442-f001:**
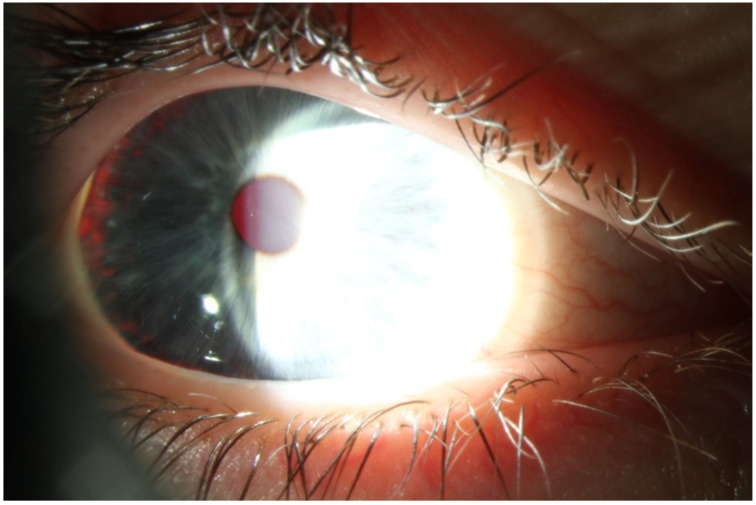
Biomicroscopy revealed blue irises with translucency grade 1 in the periphery and marked photophobia.

**Figure 2 ijms-25-06442-f002:**
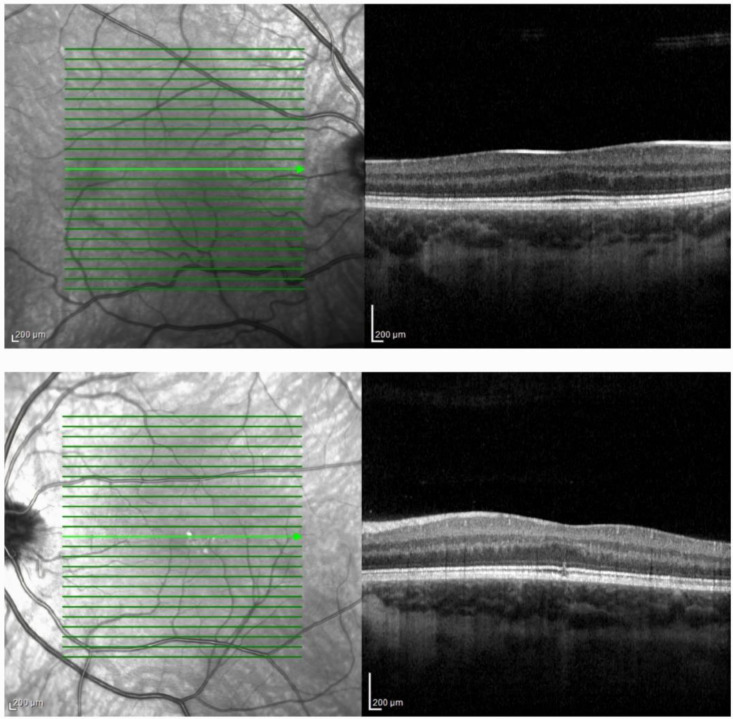
Optical coherence tomography imaging depicted flattened foveal depression on the right eye (**top image**) and the left eye (**bottom image**) with central disruption of photoreceptors and the ellipsoid zone on the left eye (**bottom image**). According to Leicester Grading System for foveal hypoplasia (FH), grade 2 FH was present on the right eye and atypical FH on the left eye.

**Figure 3 ijms-25-06442-f003:**
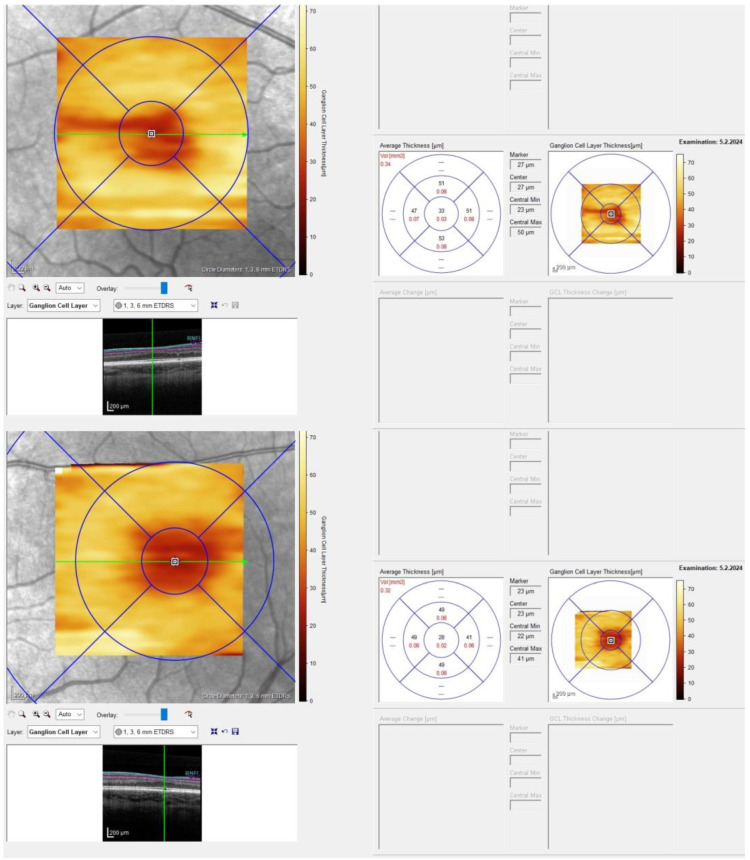
Optical coherence tomography imaging revealed similar ganglion cell layer thickness between the temporal and nasal macular areas on both eyes.

**Figure 4 ijms-25-06442-f004:**
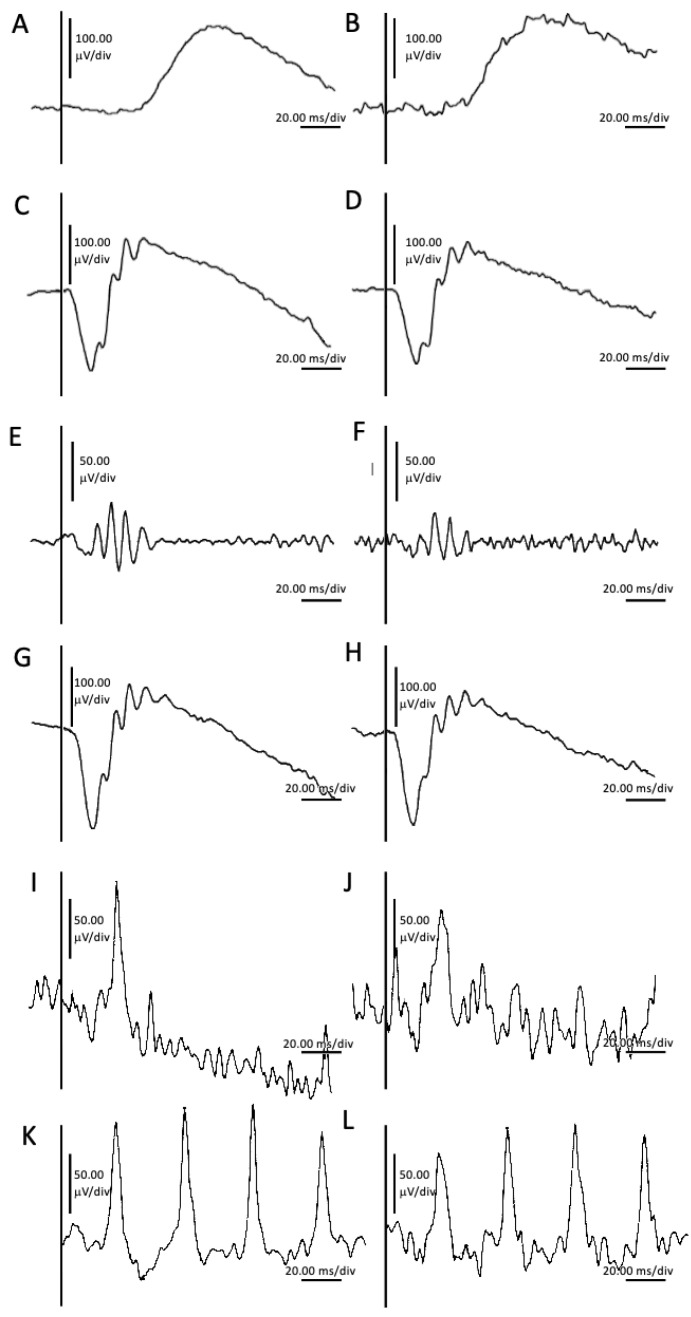
Full-field electroretinography (FFERG) testing of the right and left eye, performed according to the International Society for Clinical Electrophysiology of Vision standards, showing dark-adapted 0.01 ERG of the right eye (**A**) and left eye (**B**); dark-adapted 1.0 ERG of the right eye (**C**) and left eye (**D**); dark-adapted 3.0 oscillatory potentials of the right eye (**E**) and LE (**F**); dark-adapted 10.0 ERG of the right eye (**G**) and left eye (**H**); light-adapted 3.0 ERG of the right eye (**I**) and left eye (**J**); and light-adapted 3.0 flicker 30 Hz of the right eye (**K**) and left eye (**L**).

## Data Availability

The data presented in this case report are available on request from the corresponding author. The data are not publicity available due to privacy protection.

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
