# Peer review of "Genetic Linkage between CAPN5 and TYR Variants in the Context of Albinism and Autosomal Dominant Neovascular Inflammatory Vitreoretinopathy Absence: A Case Report"

_ijms, 2024, doi:10.3390/ijms25126442_

Round 1

Reviewer 1 Report

Comments and Suggestions for Authors

There is a concern that the CAPN5 variant is probably benign and unrelated with the patient phenotype.

Specific comments:

1) 24-27: "Moreover, synergistic effect on..." does not make sense until the pathogenicity of the variant is evident.

2) How may VUSs did you detect in the patient? You should clarify how to screen genetic variants and assess the pathogenicity in the Methods.

3) 60-68: "close chromosomal proximity underscores potential genetic and  functional relationship" needs evidence and references. Ref [9], a meeting abstract is not suitable for this discussion.

4) 74: How about family history of the patient? Family study is mandatory if a disease with autosomal dominant inheritance is suspected.

5) 236-239: "predicted to deleterious by most", how many programs suggest "pathogenic" and "begin"?

6) 309-316: again, the discussion does not make sense until the variant is considered to be pathogenic.

Author Response

Point-by-point response to reviewers

Dear Reviewer,

the authors thank you for your comments.

We have revised the manuscript entitled “Genetic linkage between CAPN5 and TYR variants in the context of albinism and autosomal dominant neovascular inflammatory vitreoretinopathy absence: a case report “ (manuscript ID: ijms-2972271) accordingly.

Authors' answers to reviewer’s comments: 

Reviewer 1:

There is a concern that the CAPN5 variant is probably benign and unrelated with the patient phenotype.

Authors' response: The authors align with the reviewer's assessment. We presume that the variant likely carries a benign nature, as elucidated within the Discussion section (lines 259-281), owing to its absence of early-onset disease presentation particularly within the framework of an already compromised and altered macula due to albinism.

Specific comments:

1) 24-27: "Moreover, synergistic effect on..." does not make sense until the pathogenicity of the variant is evident.

Authors' response:

The authors agree with the reviewers's comment.

The excerpt from lines 24-27 reads as follows: “Moreover, synergistic effect on vascular endothelial growth factor activity due to co-occurrence with pathogenic TYR and CAPN5 variants may lead to the manifestation of pronounced neovascular inflammatory vitreoretinopathy.“

In response to the reviewer's concern, we have articulated that the synergistic impact on vascular endothelial growth factor activity could result in the pronounced onset of neovascular inflammatory vitreoretinopathy, contingent upon the pathogenicity of both TYR and CAPN5 variants. It is crucial to note that this scenario does not pertain to the specific CAPN5 variant delineated in the report, which appears to exhibit a benign nature, as substantiated and discussed in lines259-281. To elucidate this distinction, we have refined the sentence to explicitly denote our consideration of any pathogenic TYR and any pathogenic CAPN5 variant: "Moreover, synergistic effect on vascular endothelial growth factor activity due to co-occurrence of pathogenic both TYR and CAPN5 variants may lead to the manifestation of pronounced neovascular inflammatory vitreoretinopathy."

2) How may VUSs did you detect in the patient? You should clarify how to screen genetic variants and assess the pathogenicity in the Methods.

Authors' response:

The authors are in accord with the reviewer's remark.

We detected heterozygous missense VUS CAPN5 c.230A>G, p.(Gln77Arg), a heterozygous missense VUS TYR c.1307G>C, p.(Gly436Ala) and a heterozygous missense variant TYR c.1205G>A, p.(Arg402Gln) which was classified as a risk factor. We added the explanation in the Methods section as requested (lines 144-158).

3) 60-68: "close chromosomal proximity underscores potential genetic and functional relationship" needs evidence and references. Ref [9], a meeting abstract is not suitable for this discussion.

Authors' response:

In response to the reviewer's comment, we have incorporated a reference to a review article discussing digenic inheritance and genetic modifiers, which elucidates the concept wherein pathogenic mutations responsible for two distinct disease entities are co-inherited, resulting in a mixed phenotype. Furthermore, we have cited a specific scenario, as presented by Morell et al., regarding the WS2-OA phenotype, which arises from digenic interaction between a gene encoding a transcription factor (MITF) and a gene that it regulates (TYR). We have omitted further explanation in the introduction section, as readers may refer to the cited articles for additional context.

Moreover, we have refined and clarified the segment to underscore that both genes play significant roles in ocular physiology and pathology, while also sharing neighboring positions on the chromosome.

„Located at q14.1, CAPN5, responsible for encoding autosomal dominant neovascular inflammatory vitreoretinopathy (ADNIV), shares this chromosome with TYR, representing a genomic neighborhood significantly implicated in ocular physiology and pathology.”

We added more references as reqested (line 65).

4) 74: How about family history of the patient? Family study is mandatory if a disease with autosomal dominant inheritance is suspected.

Authors' response:

The authors express their gratitude for the comment. Incorporated within the Case Report section, delineated within lines 83-84, is now the observation that the parents of the patient exhibit good health and normal vision. It is postulated that were they affected by ADNIV, considering their current age, they would presumably exhibit more advanced stages of the disease and would undoubtedly have manifested discernible symptoms.

5) 236-239: "predicted to deleterious by most", how many programs suggest "pathogenic" and "begin"?

Authors’ response:

The authors extend their sincere appreciation for the provided comment.

This variant is absent in gnomAD, and according to Polyphen and MutationTaster it is indicated as disease-causing and potentially damaging respectfully, while SIFT suggests it is tolerated. Consequently, the relevant section in the text has been revised accordingly (lines 259-261).

6) 309-316: again, the discussion does not make sense until the variant is considered to be pathogenic.

Authors’ response: We presume that the variant likely carries a benign nature, as elucidated within the Discussion section (lines 259-281), owing to its absence of early-onset disease presentation particularly within the framework of an already compromised and altered macula due to albinism.

In conclusion, our patient did not manifest any distinctive features specific to ADNIV. If this variant were indeed pathogenic, we would anticipate elevated VEGF levels in our patient, especially in conjunction with a pathogenic TYR variant, resulting in pronounced ADNIV. Conversely, we posit that this variant, as delineated in the discussion, is likely benign. The Discussion section has been meticulously crafted to elucidate this specific point, with every aspect carefully contextualized within the framework of a benign variant.

Thank you for your consideration of this manuscript.

Sincerely,

Mirjana Bjeloš, Ana Ćurić, Mladen Bušić, Benedict Rak, and Biljana Kuzmanović Elabjer

Reviewer 2 Report

Comments and Suggestions for Authors

This is a very well described unusual clinical picture of OCA or OA in an 11-year old boy with sharp observations and deductions of the DNA-investigations, especially of the TYR-gene.

I wondered whether the authors ascribe the subtle abnormalities in the fovea of the LE on OCT to the CAPN5-mutation? And did they check the parents for this CAPN5-mutation?

The report of the investigations of the proband is very elaborate, but I missed some general clinical information:

1) like the hair color (instead of "normal pigmentation") and whether the proband tanned normally?

2) were the low vision of the LE, the subnormal vision of the RE and the nystagmus not noticed at an early age? was he not investigated before the age of 10? or had the vision deteriorated?

3) were the parents investigated? did they have iris translucency?

4) the maternal grandmother and her brother were said to have had complete depigmentation of the skin and hair during childhood. I assume they had normal vision?

5) please indicate the visual acuity (also) in decimals or Snellen as no one uses logMAR in the clinic.

I like such case reports very much and congratulate the authors with this well written observation from which we can learn a lot.

Comments on the Quality of English Language

I think the quality of the English language is quite good. 

Author Response

Point-by-point response to reviewers

Dear Reviewer,

the authors thank you for your comments.

We have revised the manuscript entitled “Genetic linkage between CAPN5 and TYR variants in the context of albinism and autosomal dominant neovascular inflammatory vitreoretinopathy absence: a case report“ (manuscript ID: ijms-2972271) accordingly.

Authors' answers to reviewer’s comments: 

Reviewer 2:

This is a very well described unusual clinical picture of OCA or OA in an 11-year old boy with sharp observations and deductions of the DNA-investigations, especially of the TYR-gene.

I wondered whether the authors ascribe the subtle abnormalities in the fovea of the LE on OCT to the CAPN5-mutation? And did they check the parents for this CAPN5-mutation?

Author’ response:

According to the Leicester Grading System, the foveal hypoplasia here observed aligns with the classification denoted as "atypical". While existing literature has linked this type with manifestations of albinism, our review did not find evidence supporting that these alterations are markers indicative of CAPN5 mutation. Incorporated within the Case Report section, delineated within lines 83-84, is now the observation that the parents of the patient exhibit good health and normal vision. It is postulated that were they affected by ADNIV, considering their current age, they would presumably exhibit more advanced stages of the disease and would undoubtedly have manifested discernible symptoms. Genetic testing of the parents was not conducted in this study due to their documented state of health and the absence of a clinical indication warranting such testing. Furthermore, financial constraints precluded the pursuit of genetic analysis, as the associated costs would have imposed a burden on the individuals, necessitating payment for the testing.

The report of the investigations of the proband is very elaborate, but I missed some general clinical information:

1) like the hair color (instead of "normal pigmentation") and whether the proband tanned normally?

Author’ response: The patient had brown hair and tanned normally (line 85).

2) were the low vision of the LE, the subnormal vision of the RE and the nystagmus not noticed at an early age? was he not investigated before the age of 10? or had the vision deteriorated?

Author’ response: We assume that the patient displayed nystagmus and subnormal vision before undergoing initial assessment; nevertheless, these symptoms were not previously documented. Consequently, the first ophthalmological examination occurred at age 10 at a different hospital. In Croatia, visual acuity assessments are conventionally conducted at age 4 and prior to school entry, thus precluding early detection of visual deterioration. However, no indicators prompting referral for additional examinations were noted by either healthcare providers or parents.

3) were the parents investigated? did they have iris translucency?

Author’ response: We did not examine the parents as they were healthy and did not have any symptoms.

4) the maternal grandmother and her brother were said to have had complete depigmentation of the skin and hair during childhood. I assume they had normal vision?

Author’ response: The maternal grandmother and her brother reportedly exhibited complete depigmentation of the skin and hair during childhood. Information regarding their visual acuity during that period is unavailable due to the less advanced state of the healthcare system at that time. However, according to the recollection of the mother, no family member experienced significant visual impairment.

5) please indicate the visual acuity (also) in decimals or Snellen as no one uses logMAR in the clinic.

Author’ response: We added decimals as requested (lines 85-91).

Thank you for your consideration of this manuscript.

Sincerely,

Mirjana Bjeloš, Ana Ćurić, Mladen Bušić, Benedict Rak, and Biljana Kuzmanović Elabjer

Round 2

Reviewer 1 Report

Comments and Suggestions for Authors

The revised version of the manuscript shows that using a panel sequence system, Blue-145 print Genetics Retinal Dystrophy Panel Plus, they found three mentioned variants. As the CAPN5 variant was not confirmed to be pathogenic, or the patient was not associated with ADNIV, the manuscript should be revised further:

1) Please clarify if there were no additional VUSs in the panel test.

ï¼’) 24-27: “Moreover, synergistic effect on .." should be revised as the synergistic effect is not fully evident based on the PubMed.

3) 60-68: ''close chromosomal proximity underscores potential genetic and functional relationship”; TYR is in 11q14.3, CAPN5 in 11q13.5. They are ~10Mb apart from each other. The phrase should be revised.

ï¼”) 236-239: "predicted to deleterious by most"; The authors should address more in-silico programs for evaluating the pathogenicity impartially (I have checked http://varsom.com).

5) It is better to remove the sentence, “The absence of this phenomenon in our case supports the probable benign nature of the CAPN5 variant” from the Abstract (it is still confusing).

Author Response

Dear Reviewer,

the authors thank you for your comments.

We have revised the manuscript entitled “Genetic linkage between CAPN5 and TYR variants in the context of albinism and autosomal dominant neovascular inflammatory vitreoretinopathy absence: a case report “ (manuscript ID: ijms-2972271) accordingly.

Authors' answers to reviewer’s comments: 

Reviewer 1:

The revised version of the manuscript shows that using a panel sequence system, Blue-145 print Genetics Retinal Dystrophy Panel Plus, they found three mentioned variants. As the CAPN5 variant was not confirmed to be pathogenic, or the patient was not associated with ADNIV, the manuscript should be revised further:

1) Please clarify if there were no additional VUSs in the panel test.

Authors' response: There were no additional VUSs in the panel. Only the aforementioned variants were identified.

ï¼’) 24-27: “Moreover, synergistic effect on .." should be revised as the synergistic effect is not fully evident based on the PubMed.

Authors' response: The authors agree with the reviewer’s comment. The sentence was revised accordingly (lines 24-25).

3) 60-68: ''close chromosomal proximity underscores potential genetic and functional relationship”; TYR is in 11q14.3, CAPN5 in 11q13.5. They are ~10Mb apart from each other. The phrase should be revised.

Authors' response: The authors are in accord with the reviewer's remark. The sentence was revised as requested (lines 58-62).

ï¼”) 236-239: "predicted to deleterious by most"; The authors should address more in-silico programs for evaluating the pathogenicity impartially (I have checked http://varsom.com).

Authors' response: More in-silico programs were added as requested (lines 245-249).

5) It is better to remove the sentence, “The absence of this phenomenon in our case supports the probable benign nature of the CAPN5 variant” from the Abstract (it is still confusing).

Authors' response: The sentence was omitted from the Abstract and Conclusion, as requested.

Thank you for your consideration of this manuscript.

Sincerely,

Mirjana Bjeloš, Ana Ćurić, Mladen Bušić, Benedict Rak, and Biljana Kuzmanović Elabjer

Round 3

Reviewer 1 Report

Comments and Suggestions for Authors

The authors further revised the manuscript for the reviewer's comments.  I agree with the revision. 

Author Response

The authors thank the Reviewer for the comments.